# DBSCAN and GIE, Two Density-Based “Grid-Free” Methods for Finding Areas of Endemism: A Case Study of Flea Beetles (Coleoptera, Chrysomelidae) in the Afrotropical Region

**DOI:** 10.3390/insects12121115

**Published:** 2021-12-13

**Authors:** Maurizio Biondi, Paola D’Alessandro, Walter De Simone, Mattia Iannella

**Affiliations:** Department of Life, Health & Environmental Sciences, University of L’Aquila, Via Vetoio Coppito, 67100 L’Aquila, Italy; maurizio.biondi@univaq.it (M.B.); paola.dalessandro@univaq.it (P.D.); walter.desimone@graduate.univaq.it (W.D.S.)

**Keywords:** areas of endemism, density-based clustering, DBSCAN, GIE, Chrysomelidae, Afrotropical region, GIS analysis, ArcGIS Pro, Model Builder

## Abstract

**Simple Summary:**

Areas of endemism (AoEs) are one of the most important topics discussed in biogeography, considering that the analysis of areas of sympatry between endemic species is essential to understand species distribution patterns, reconstruct evolutionary events, regionalize biogeographical areas, and assess regions of high conservation concern. Here, we propose a workflow based on the application of a clustering-based algorithm to identify AoEs and compare it to another method, the Geographical Interpolation of Endemism, based on a kernel density approach. We apply this framework to the flea beetles of the whole sub-Saharan Africa, identifying several AoEs through both methods, but with differences in their delimitation, number and features of characteristic species, and surface. Considering that our proposed workflow can be applied to any territorial context and sets of endemic species, we also provide a GIS tool that implements all the steps into one single toolbox. The identification of AoEs, possibly facilitated by our approach, can provide useful spatial information when dealing with several biodiversity-related issues, even applied to practical conservation measures, such as protected areas management and landscape planning.

**Abstract:**

Areas of endemism (AoEs) are a central area of research in biogeography. Different methods have been proposed for their identification in the literature. In this paper, a “grid-free” method based on the “Density-based spatial clustering of applications with noise” (DBSCAN) is here used for the first time to locate areas of endemism for species belonging to the beetle tribe Chrysomelidae, Galerucinae, Alticini in the Afrotropical Region. The DBSCAN is compared with the “Geographic Interpolation of Endemism” (GIE), another “grid-free” method based on a kernel density approach. DBSCAN and GIE both return largely overlapping results, detecting the same geographical locations for the AoEs, but with different delimitations, surfaces, and number of detected sinendemisms. The consensus maps obtained by GIE are in general less clearly delimited than the maps obtained by DBSCAN, but nevertheless allow us to evaluate the core of the AoEs more precisely, representing of the percentage levels of the overlap of the centroids. DBSCAN, on the other hand, appears to be faster and more sensitive in identifying the AoEs. To facilitate implementing the delimitation of the AoEs through the procedure proposed by us, a new tool named “CLUENDA” (specifically developed is in GIS environment) is also made available.

## 1. Introduction

Endemisms are one of the most important features in the distribution of biodiversity on Earth, and their identification is essential to define the biological value of an area and its intrinsic conservation requirements [1,2,3]. The endemicity of a species is a result of both ecological and historical factors [4,5]; while ecology explains how biotic and abiotic factors can shape a species’ range, a historical reconstruction can uncover which geological and evolutionary events contributed to confine a species to its current distribution [6]. Areas of endemism are a widely explored concept in biogeography, as they are identified by the spatial overlapping of two or more endemic species [7,8,9,10,11]. In fact, a non-random distributional congruence between taxa can identify common evolutionary biogeographical processes [12]. Throughout geological time, an assemblage of endemic species sharing a common space might have responded differently to the same ecological factors: this is the reason why each area of endemism usually has fuzzy edges [11,13], making it more difficult for biogeographers to define its exact borders.

There is still a high level of disagreement over what the areas of endemism actually represent, and what the correct way to identify them is [7,14]. Despite this, all authors agree that these spatial units are dynamic entities, representing a current snapshot of the evolution of species, or groups of species, sharing a common history [15]. Therefore, it is essential to identify areas of endemism not only to infer the history of biogeographical units [16,17], but also to lay the groundwork for suitable conservation plans within a specific study area [3,18]. Different methods have been proposed to detect areas of endemism: Parsimony Analysis of Endemicity (PAE) [8,19,20], Cladistic Analysis of Distributions and Endemism (CADE) [21], Endemicity Analysis with Optimality Criterion [22,23,24], and Network Analysis [25] are only a few examples of the numerous techniques suggested throughout the years to identify and analyse these historical and ecological units.

In this paper, two different approaches to infer areas of endemism independent of grid cells (“grid-free”) are compared and discussed: the Density-Based Spatial Clustering of Application with Noise (DBSCAN), applied here for the first time to identify areas of endemism, and the Geographic Interpolation of Endemism (GIE), recently proposed by Oliveira et al. [26]. Both these methods, instead of dividing the study area into grid cells, use the species’ distribution points as raw data to detect areas of endemism. DBSCAN, proposed by Ester et al. [27], is a density-based clustering algorithm that works on the assumption that clusters are higher-density regions in space separated by regions of lower density (noise). The DBSCAN measure of density uses the distances between data points and applies local or global density criteria to separate out clusters from noises, represented by the not allocated points. This approach does not rely on a pre-defined number of clusters, such as the several variants of K-Means Clustering or Fuzzy C-means and can be made flexible for different shapes and varying densities of points.

GIE, instead, is based on a modified kernel interpolation function [26]. The kernel estimator has been widely used in ecology to estimate the density of species occurrence and consequently draw species range probability maps [28]. With this method, an area of influence is defined for each species range, and its radius is used to categorize species into different classes sharing areas of influence of similar size. A Gaussian function is applied for each occurrence, obtaining a homogeneous circular area of influence in which the function values decrease when moving away from the occurrence record (which has the highest value). Then, a kernel density function (an algorithm that expresses the spatial density of features in a geographic space) calculates the occurrence of endemism by identifying the overlap between different areas of influence, weighted by the degree of overlapping [29].

DBSCAN and GIE are here applied to a database of flea beetle species (Chrysomelidae, Galerucinae, Alticini) endemic to the Afrotropical region. Due to their high number of species and habitat specificity, often entailing a restricted geographic distribution, the insects are generally considered adequate candidates for biogeographical studies on endemism [30]. The leaf beetles (Coleoptera Chrysomelidae) represent a large portion of the herbivorous insect fauna for many habitats [31,32,33] and are considered a useful tool to analyse insect community structure [34,35,36]. In particular, the Alticini are a tribe of Chrysomelidae comprising over 534 genera and about 8000 species [37,38], occurring all over the world. Members of this tribe are commonly defined as flea beetles because of the presence of a metafemoral extensor tendon that enables them to jump [39,40]. Adult and larval stages mainly feed on stems, leaves, or roots, although rarely on flowers, of almost all the higher plant families, generally with high levels of specialization and in different environments [31,41,42,43]. The Afrotropical flea beetle fauna includes about 1600 known species in 103 genera, of which over 80% are endemic to sub-Saharan Africa and/or Madagascar [44,45,46].

## 2. Materials and Methods

### 2.1. Study Area and Species Database

The study area (Figure 1) comprises the Afrotropical region as defined by Udvardy and Udvardy [47]. The analyses were performed on a dataset including 337 well-known species of flea beetles (Chrysomelidae, Galerucinae, Alticini) occurring in the Afrotropical region, with a total of 3296 records of presence. Distribution data (database available from the authors upon request) were collected from peer-reviewed literature (Appendix A), and the species identification of every specimen was confirmed by the authors (MB and PD). Geographic coordinates possibly missing were retrieved using Google Earth.

### 2.2. Abbreviations Used

AMM: Amber Mountain; ANR: Antananarivo region; AoE: area of endemism; BER: Betsiboka region; DBSCAN: Density-Based Spatial Clustering of Application with Noise; dc: maximum distance between the centroids; dd: maximum diameter of the distribution of the species; DKM: Drakensberg Mountains; GIE: Geographical Interpolation of Endemism; KAR: Katanga region; KIL: Kilimanjaro region; KLR: Kivu Lake region; KWN: KwaZulu-Natal; LI: Limpopo; MP: Mpumalanga; WCP: Western Cape Province.

### 2.3. Geographical Interpolation of Endemism (GIE)

This analysis was performed using the tool “GIE” in the BioDinamica package [48]. The maximum value of centroid distance was chosen as a parameter to define species classes. To compare the results of GIE with those of DBSCAN, the species database was divided into 3 classes, with the first class (Class 1) having a radius of the distance between centroids equal to or less than 100 km, the second one (Class 2) equal to or less than 150 km, and the third one (Class 3) equal to or less than 200 km. The analyses of the three classes considered (Class 1, Class 2, and Class 3) were performed separately due to unclear results obtained whether a single analysis including the three classes altogether is performed.

### 2.4. Density-Based Spatial Clustering of Application with Noise (DBSCAN)

As also stated in the Introduction, it is important to highlight that other existing clustering methods, such as K-means and its variants, or Fuzzy C-means, require setting an a priori number of clusters to carry out the analysis. Thus, these algorithms are not indicated in our case, considering that the number of endemic areas should be a result of an appropriate analysis, and not a parameter pre-determined by the operator.

The DBSCAN algorithm was preferred to HDBSCAN (Hierarchical Density-Based Spatial Clustering of Applications with Noise) [49], because this last one does not select clusters based on a global “epsilon” threshold (distance between centroids in our case) but creates a hierarchy for all possible epsilon values and thus only requires “minPts” (number of sinendemisms in our case) as single input parameter. The epsilon parameter used by DBSCAN is instead useful to discover all the clusters (AoEs) of variable densities with a definite value of epsilon (distance between centroids).

This analysis was performed using the tool Density-based Clustering implemented in the Spatial Statistics toolset of ArcGIS Pro 2.8. For ease of reference, in the workflow reported after, we name as “sp” (species name), “x” (longitude), and “y” (latitude) the fields of the input table hosting the species’ occurrence data; during the steps proposed, dummy names also will be assigned to intermediate files.

Step 1: Implementation of the dataset

Set the fields of a table in the order: sp, x, y; longitude and latitude are reported in decimal degrees. Then, load and display the occurrences (points) from the table in GIS environment.

Step 2: Selection of species

Use, as the first algorithm, the tool “Minimum Bounding Geometry” in ArcGIS Pro. It is necessary to implement the whole process, setting Geometry type = Circle, Group option = List and Group Field = sp (to group the output of this tool based on the species); furthermore, check the “Add geometry characteristics as attributes to output”. Then, the output features with a Minimum Bounding Geometry value (MBG_Diameter) equal to or less than the chosen diameter (in our case: ≤1.00 for species with a maximum distribution range width equal to or smaller than 100 km; ≤3.00 for species at 300 km; ≤5.00 for species at 500 km) were selected and exported to new files, named, for instance, “species_100 km”, “species_300 km”, or “species_500 km”, respectively. This step permits creating three selections of species with a distribution range width limited to 100, 300, or 500 km, respectively, avoiding the chance of including species with broad or disjunct distributions, which could cause a disturbing factor in the analysis.

Step 3: Calculation of the centroids

Calculate the centroids of the files obtained through the previous distribution range filtering (i.e., “species_100 km”) and convert this information, together with the corresponding species name, to a point-geometry file named, for instance, “centroids_100 km”. Apply the Density-based Clustering tool, using the different distance between centroids (“Defined Distance” = 100, 150, 200 km, similar to the distances used in GIE), with the option “Minimum Features” ≥2, depending on the number of sinendemisms one wants to consider (five in our case). This parameter could also be used “backwards”, starting from a high number of sinendemic species to infer the highest number of sinendemisms by which the areas of endemism start to form. Then, use the Intersect tool between the DBSCAN file (i.e., DBSCAN_100 km_50 km_5 sin) and the corresponding centroid data used for the analysis (i.e., centroids_100 km), to obtain the information about the names of the species included in the clusters identified by DBSCAN.

Step 4: Developing Areas of endemism

As the last step, after having to discard in the attribute table all records reporting the −1 value (noise species not allocated into a specific cluster), use the tool “Create buffer” to definite the areas of endemism around the clusters obtained by the DBSCAN, setting the following parameters: Buffer Type = Distance, Buffer Distance = 80 km (in our case, but this distance is depending by the scale of the referring area), Dissolve Option = List, Dissolve Fields = Cluster ID (to define the buffers based on the clusters).

### 2.5. The CLUENDA Tool

To implement the identification of areas of endemism using DBSCAN as described above, a new tool was developed in GIS environment through the Model Builder of ArcGIS Pro. The tool, named CLUENDA (Figure 2 and Appendix A), aims to speed up data processing and ensure the comparability of maps of endemism areas obtained from species databases. The framework combines several geoprocessing tools in the ArcGIS Pro 2.8 environment to generate the necessary factors to identify the areas based on the clusters of sinendemisms. ArcGIS Pro Model Builder combines several GIS operations and runs these modules on a single spatial dataset (point features). Each model consists of three fundamental elements: the input parameters, the geoprocessing tools, and the output data. Model parameters are model-specific inputs that must be user-defined (e.g., species occurrences, to which a geographic projection should be applied; instead, chordal distances will be used). The geoprocessing tools produce output data in a defined sequence using the input datasets until areas with clustered sinendemic species are obtained. With the CLUENDA tool, we have mapped the areas of endemism inferred based on: (a) three different selections of species with maximum diameter of the distance to 100, 300 and 500 km, respectively; (b) three different distances between centroids up to 100, 150, and 200 km, respectively.

To investigate possible differences in computation performances, execution times were measured for both GIE and CLUENDA. A 4-core Intel i7 processor (2.80 GHz), equipped with 32 GB of RAM (Windows 10) was used for this purpose.

The analyses were run with the dataset described above, as well as with a 10-times-bigger dummy dataset (generated based on the original dataset used, thus containing 32,960 points for the whole sub-Saharan Africa) to stress-test the two software used.

## 3. Results

### 3.1. Areas of Endemism (AoEs)

The execution times needed for the identification of the AoEs for the “regular” dataset were 75 s for GIE and 5 s for CLUENDA; the dummy “10-times bigger” dataset took 96 s and 6 s for the computation in GIE and CLUENDA, respectively.

The AoEs identified by both GIE and DBSCAN are listed below in alphabetical order:-AMM (Amber Mountain): this refers to the Amber Mountain, a famous protected area in the Diana region of Northern Madagascar. AMM is well known for its endemic flora and fauna. AMM is a part of the “Madagascar and the Indian Ocean Islands” biodiversity hotspot [50].-ANR (Antananarivo region): this AoE is located in Central Madagascar and mainly includes Anamalanga, Bongolava, Itasy, and Vakinankaratra regions. ANR is a part of the “Madagascar and the Indian Ocean Islands” biodiversity hotspot.-BER (Betsiboka region): this AoE is located in Northern Madagascar and includes, in addition to the Betsiboka, the Sofia and Analanjirofo regions. BER is a part of the “Madagascar and the Indian Ocean Islands” biodiversity hotspot.-DKM-KWN (Drakensberg Mountains–KwaZulu Natal): this AoE comprehends the Drakensberg Mountains, which are main mountain range in southern Africa, and the coastal areas of KwaZulu Natal. The Drakensberg Mountains rise to more than 3475 m a.s.l., extending roughly from northeast to southwest for over 1100 km parallel to the southeastern coast of South Africa. They are part of the Great Escarpment and separate the extensive highlands of the South African interior from the lower lands along the coast. In the Drakensberg Mountains, alpine grasslands and small pockets of Afromontane Forest are present. The coastal regions of KwaZulu-Natal typically have subtropical thickets and deeper ravines; steep slopes host small spots of Afromontane Forest. The midlands have moist grasslands. The northern area has a primarily moist savannah habitat. DKM-KWN is a part of the “Maputaland–Pondoland–Albany” biodiversity hotspot.-KAR (Katanga region): this AoE is located in the Democratic Republic of Congo and mainly includes the ridges of the plateaus of Katanga (Shaba) province. They include Kundelungu (1600 m a.s.l.), Mitumba (1500 m), and Hakansson (1100 m) mountains. The Katanga plateaus reach as far north as the Lukuga River and contain the Manika Plateau, the Kibara and the Bia mountains, and the high plains of Marungu. Despite the high plant diversity present [51], KAR is not included in any biodiversity hotspot.-KIL (Kilimanjaro region): this AoE has the Kilimanjaro region as its central area. Towards the southeast, it extends to the Tsavo West National Park in Kenya and Mkomazi National Park in Tanzania, while northwards it extends to the Amboseli National Park. KIL comprehends both montane and savannah areas. The Kilimanjaro area is a part of the “Eastern Afromontane” biodiversity hotspot.-KLR (Kivu Lake region): this AoE is located in the area of the Kivu Lake in the Albertine Rift Valley and also includes the mountain areas of Birunga, Volcan Mikeno, and Volcan Karisimbi, between Uganda, Rwanda, and the Democratic Republic of Congo (Kivu Sud). KLR is a part of the “Eastern Afromontane” biodiversity hotspot.-MP-LI (Mpumalanga–Limpopo): this AoE comprehends part of the Mpumalanga and Limpopo provinces. Mpumalanga is divided by the Drakensberg escarpment into a westerly half consisting mainly of high-altitude grassland called the Highveld and an eastern half situated in low-altitude subtropical Lowveld/Bushveld, mostly savannah habitat. The Lowveld is relatively flat with interspersed rocky outcrops. The Lebombo Mountains form a low range in the far east, on the border with Mozambique. Limpopo contains much of the Waterberg Biosphere, a massif of approximately 15,000 km^2^ which is the first region in the northern part of South Africa to be named a UNESCO Biosphere Reserve (http://waterbergbiospherereserve.org/why-the-waterberg-is-a-biosphere.html, accessed on 12 October 2021). MP-LI is partially included in the “Maputaland–Pondoland–Albany” biodiversity hotspot.-WCP (Western Cape Province): this AoE is mainly restricted to South Africa’s Western Cape Province. Most of the region is covered with fynbos, a sclerophyllous shrubland occurring on acid sands or nutrient-poor soils derived from Table Mountain sandstones. This area covers the Mediterranean climate region of South Africa in the southwestern corner of the country and extends eastward into the Eastern Cape Province. WCP is a part of one of the world’s six floral kingdoms, the “Cape Floristic Region” biodiversity hotspot.

### 3.2. GIE

The weighted consensus map obtained by the GIE analysis, setting the “Minimum number of endemism” = 5, is shown in Figure 3a. In this map, kernels identify seven AoEs, five larger—WCP (32 sinendemisms), DKM-KWN (19), MP-LI (14), and KIL (12) in sub-Saharan Africa and AMM+BER+ANR (51) in Madagascar—and two smaller—KAR (7) and KLR (5) in Central Africa. There is no significant difference in the results between the weighted and the unweighted maps.

As the intrinsic value of an endemic species is weighted by the width of its distribution range, the kernel maps of the three classes selected (Class 1, Class 2, Class3) are displayed individually (Figure 3b–d), and the endemic species included in each of them are listed in Appendix A. The use of different classes was performed to identify areas of endemism with a higher degree of specificity based on the different distances of the centroids of the species range. In Class 1, which includes only the species whose distances between the centroids is equal to or smaller than 100 km, only small and fragmented areas were identified and attributed to the main five AoEs (Figure 3b; Table 1): western part of WCP (20 sinendemisms), MP-LI (13), and KAR (6) for sub-Saharan Africa and AMM (6) and BER + ANR (28) for Madagascar. Instead, considering the species with distances between the centroids equal to or smaller than 150 km (Class 2), the seven AoEs identified were located (Figure 3c; Table 1): in WCP (+11 sinendemisms), BER + ANR (+8), with larger areas, and in MP-LI, KAR, and AMM, unchanged; to these the areas DKM-KWN (17) and KLR (5) were added for sub-Saharan Africa. Finally, with distances between the centroids equal to or smaller than 200 km (Class 3), the definitive seven AoE (Figure 3d; Table 1) were detected similarly to those identified in the consensus maps. The correlation values among the maps were calculated for the three classes considered. The highest value (*r* = 0.92525) is between Class 2 and Class 3, while the lesser (*r* = 0.73043) is between Class 1 and Class 3.

### 3.3. DBSCAN

The results by DBSCAN analysis for identifying AoEs, setting the number of sinendemisms to 5, are reported in Figure 4, Figure 5, Figure 6, Figure 7, Figure 8 and Figure 9, and the endemic species included in each of them are listed in Appendix A. The extension, the number of sinendemisms, and the ratio “number of endemic species/total of species” for each area are reported in Table 1 and Table 2 and in Figure 10.

In general, it can be observed that as the distribution interval of the species considered increases, respectively, with a maximum diameter of 100, 300, and 500 km, the identified AoEs are, at the same distance between the centroids, homogeneous in their geographical location, although they may be more extensive or merged into larger areas (Figure 4, Figure 5 and Figure 6).

Considering 100 km as the maximum distance between the centroids of the selected species (“defined distance” = 100 km), with the species with distribution up to 100 km (150 species out of 337 total), the analysis identifies the following seven AoEs (Figure 4, Table 1): KAR (6 sinendemisms) in Central Africa; DKM-KWN (10), MP-LI (11), and WCP (21) in Southern Africa; AMM (5), BER (11), and ANR (10) in Madagascar. With species up to 300 (181 out of 337) and 500 km (197 out of 337) (Table 1), a general increase in surface is observed for the DKM-KWN (+3 sinendemisms) and WCP (+11) areas, as well as a merger of the BER and ANR areas (35). On the other side, the areas KAR, MP-LI, and AMM remain unchanged. By increasing the “defined distance” in DBSCAN to 150 and 200 km, the results are quite different. In the first case, the analysis identifies two new AoEs in Eastern Africa—KIL, with 8, 11, and 17 sinedemisms, respectively, and KLR, with 5, 7, and 7 sinendemisms—while an area increase is observed for DKM-KWN and MP-LI, which merge into a unique area when the range of the species considered is up to 500 km (Figure 5, Table 1). By increasing the distance of the centroids to 200 km (“defined distance” = 200 km), in all three cases (species up to 100, 300, and 500 km), we observe the merger of the DKM-KWN and MP-LI areas and the gradual increase, in terms of surface, of KLR and KAR (Figure 6, Table 1). The correlation values (Kendall index) among the maps obtained by the different combinations of the widths of the distribution ranges and the distance between centroids are reported in Table 3. The correlation index decreases quickly as the distance between the centroids increases (100, 150, 200 km) rather than with the increase in the width of the distribution ranges (100, 300, 500 km). The non-parametric Kendall index was preferred because it uses pairs of observations and determines the strength of association based on the pattern of concordance and discordance between the pairs, which best fits our analysis.

## 4. Discussion

DBSCAN and GIE both return largely overlapping results, detecting the same geographical locations for the AoEs, even if their results as a whole result in generally different delimitation and surface and the number of the detected sinendemisms (Table 2), also resulting in very different computation times. The consensus maps obtained by GIE are in general less clearly delimited than the maps obtained by DBSCAN, but nevertheless allow us to evaluate the core of the AoEs more precisely, as they can be displayed with a “stretched” symbology representing of the percentage levels of the overlap of the centroids, keeping a sort of hierarchy between the nested areas. DBSCAN, on the other hand, appears to be more sensitive in identifying areas of endemism. This method, in fact, detects the same areas as GIE but with comparatively shorter distances between the centroids. For example, with a “defined distance” of 150 km, regardless of the extent of the distribution of the species used in the analysis (100, 300, or 500 km in diameter), DBSCAN identifies the position of all the possible areas, including KIL and KLR, while with GIE, the KLR area is identified only in Class 2 (radius 150 km) and the KIL area in Class 3 (200 km). The same is also true for the DKM-KWN area, identified by DBSCAN with a distance between centroids of 100 km and not by GIE. Creating datasets of species with different widths of the distribution ranges, such as those up to 100, 300, and 500 km here proposed, allows DBSCAN to identify more precise and better-defined areas, with a higher number of identified sinendemisms (Table 2). Generally, in DBSCAN, the increase in the width of the distribution range of the species selected for the analysis does not allow the identification of new AoEs or the disappearance of them, but only larger areas, or the merge of areas previously identified using species with more limited distributions. New areas are detected only with the increase in the distance between centroids.

From the conservation/landscape planning point of view, it is important to highlight the main difference between the two approaches: the DBSCAN produces discrete territorial units, while GIE gives a continuous output. Researchers, stakeholders, and decision-makers should bear this in mind when searching for AoEs to manage or protect them. For instance, if GIE is applied and a specific territory has to be managed, a threshold value must be chosen to discretize the GIE’s output. This may introduce subjectivity, but also strengthen the results if this choice is supervised by experts. On the other hand, DBSCAN was allowed to identify comparable AoEs with shorter distances between centroids (with respect to GIE), also detecting a higher number of sinendemisms for each of them. Thus, the application of DBSCAN may be suggested when more conservative solutions are needed.

## 5. Conclusions

GIE and DBSCAN consider the degree of overlap between species ranges as a pivotal parameter. This requirement stems from the necessity to bypass the geometric constraints and spatial scale bias imposed by grid cell use. In fact, some authors have previously argued that areas of endemism should be modelled by using more “natural” geographical units instead of overlapping geometric-sized grid cells onto the study area [52]. Additionally, areas of endemism should have irregular edges, and quadrats traditionally used in methods grid-based commonly fail to fully describe the dynamic structure of species distributions [15]. Therefore, the methodologies here discussed are a promising way to implement the research of areas of endemism by taking into account all the aforementioned observations.

## Figures and Tables

**Figure 1 insects-12-01115-f001:**
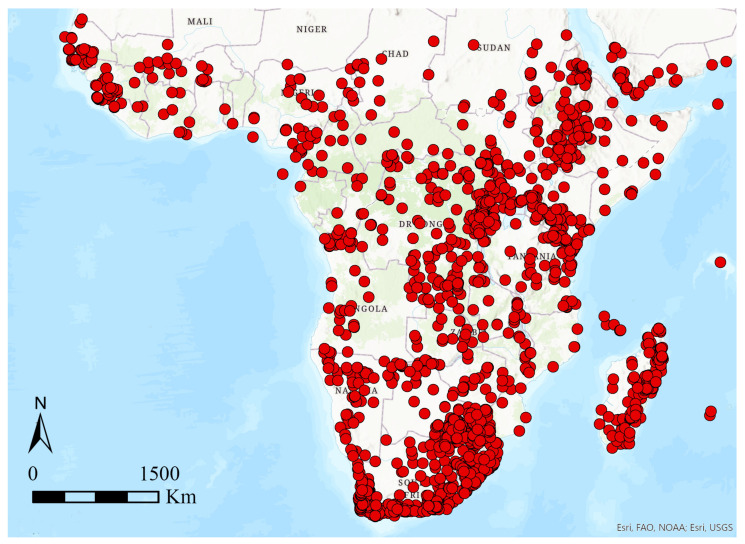
Study area and occurrence localities used for analysis (see text).

**Figure 2 insects-12-01115-f002:**
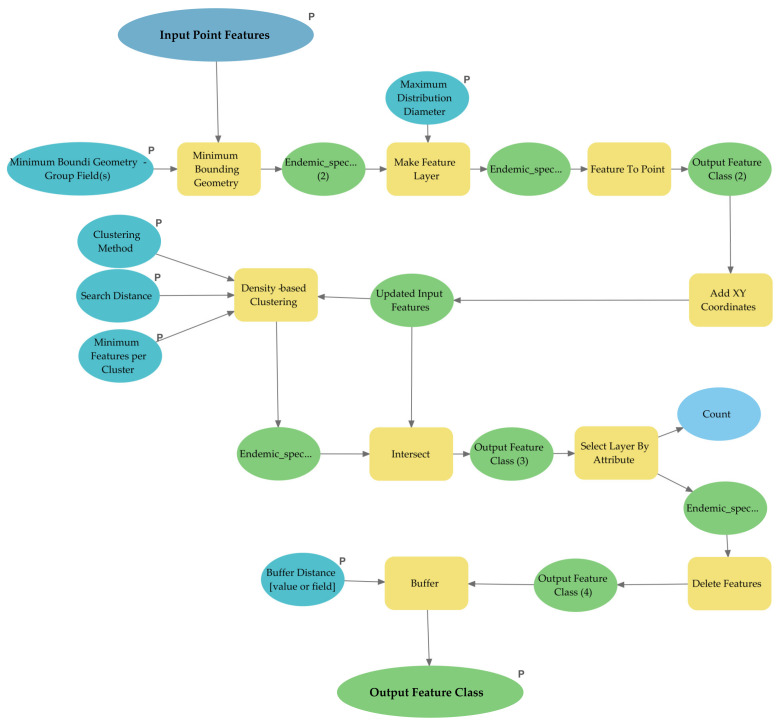
Scheme of the CLUENDA toolbox, developed in ArcGIS Pro 2.8.

**Figure 3 insects-12-01115-f003:**
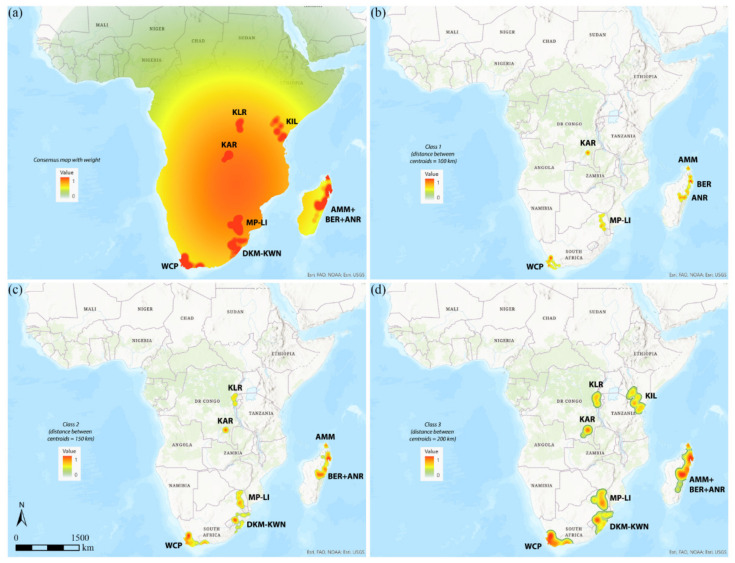
(**a**) Consensus maps of the areas of endemism detected by GIE. (**b**) Areas of endemism detected by GIE in Class 1 (100 km), (**c**) Class 2 (150 km) and (**d**) Class 3 (200 km).

**Figure 4 insects-12-01115-f004:**
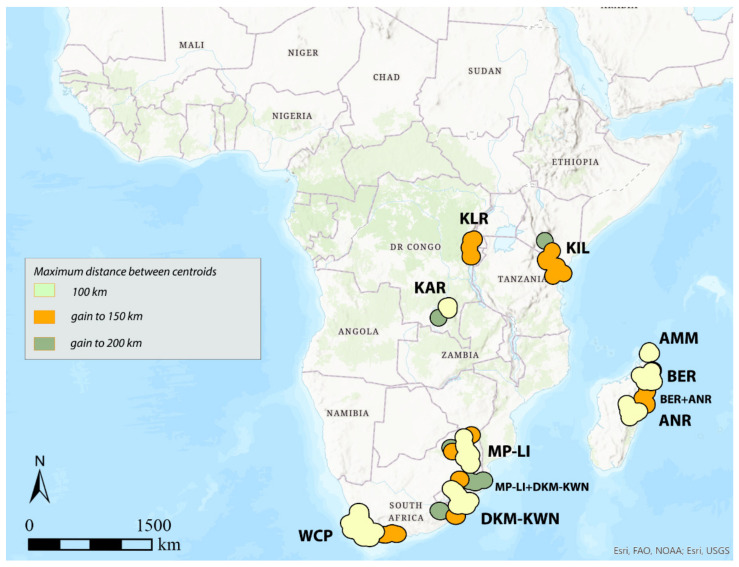
Areas of endemism detected by DBSCAN using species with distribution range up to 100 km and distance between centroids of 100, 150, and 200 km.

**Figure 5 insects-12-01115-f005:**
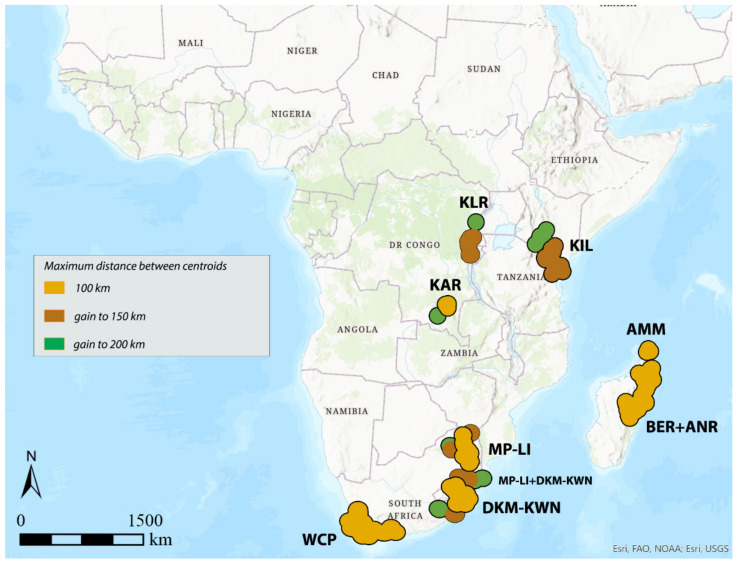
Areas of endemism detected by DBSCAN using species with distribution range up to 300 km and distance between centroids to 100, 150, and 200 km.

**Figure 6 insects-12-01115-f006:**
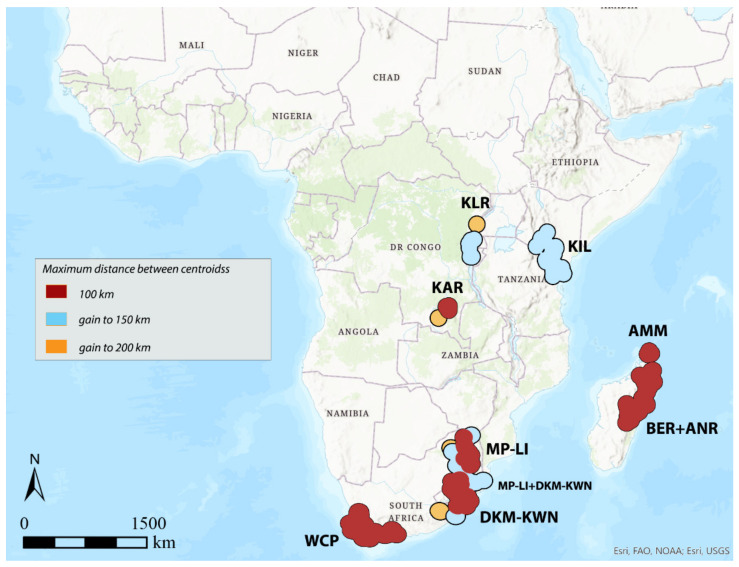
Areas of endemism detected by DBSCAN using species with distribution range up to 500 km and distance between centroids to 100, 150, and 200 km.

**Figure 7 insects-12-01115-f007:**
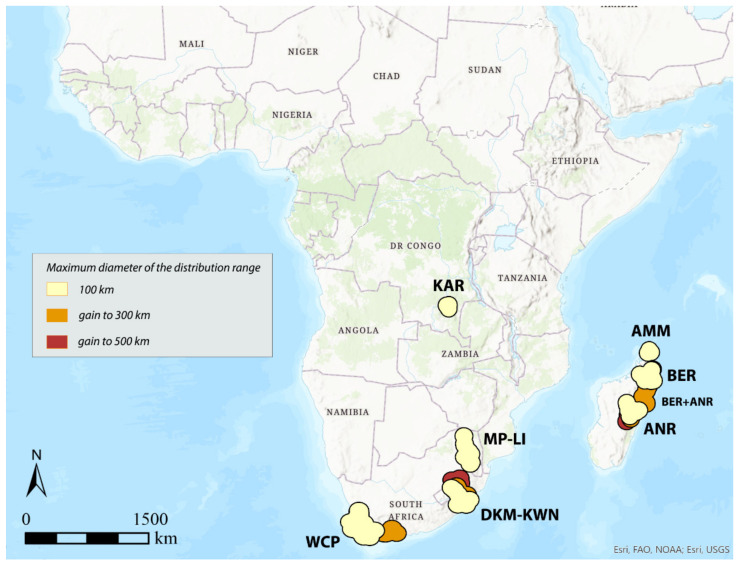
Areas of endemism detected by DBSCAN using the distance between centroids to 100 km and species with distribution range up to 100, 300, and 500 km.

**Figure 8 insects-12-01115-f008:**
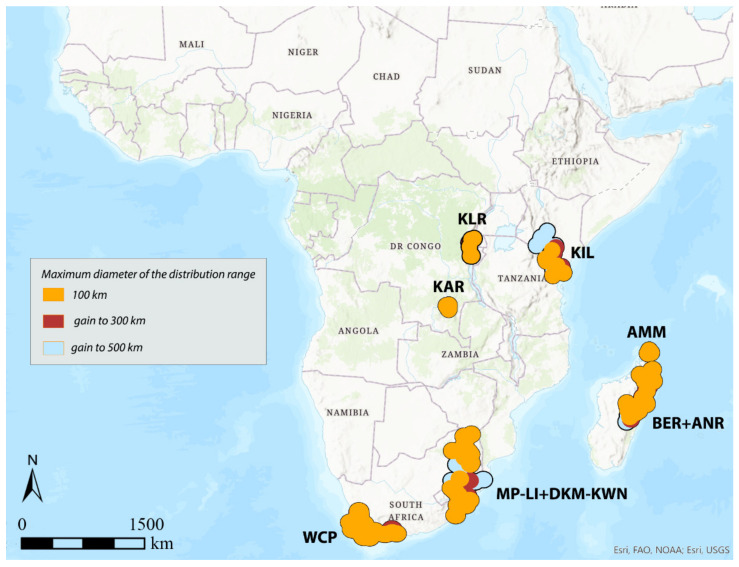
Areas of endemism detected by DBSCAN using the distance between centroids to 150 km and species with distribution range up to 100, 300, and 500 km.

**Figure 9 insects-12-01115-f009:**
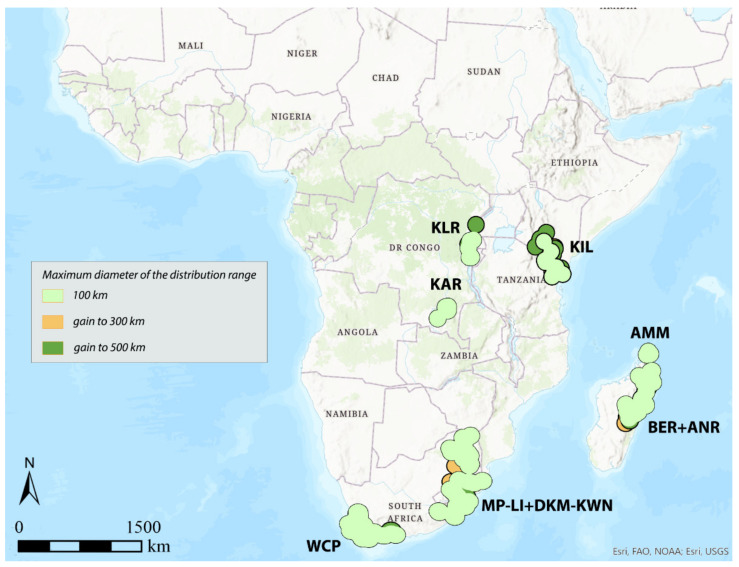
Areas of endemism detected by DBSCAN using the distance between centroids to 200 km and species with distribution range up to 100, 300, and 500 km.

**Figure 10 insects-12-01115-f010:**
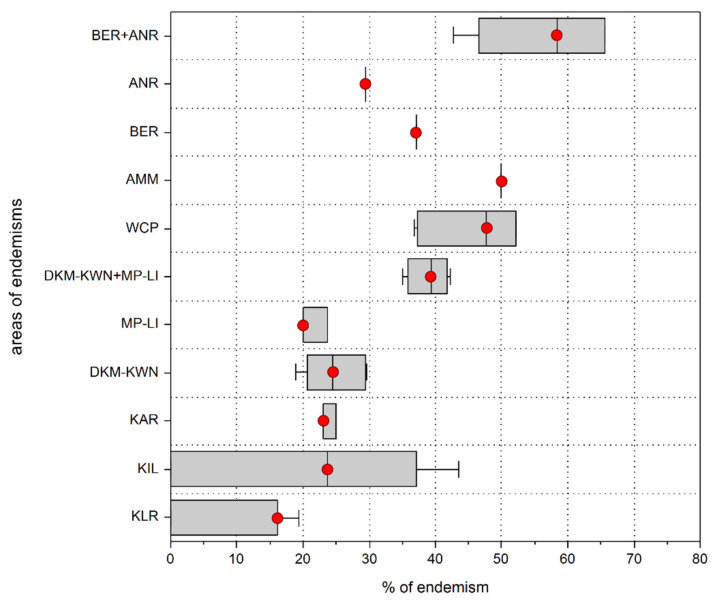
Box plots of the ratio “number of endemic species/total of species” for each of the AoEs identified by the DBSCAN analysis. Red spot = median.

**Table 1 insects-12-01115-t001:** Surface and number of sinendemisms for the AoEs identified by the DBSCAN analysis. dd = maximum diameter of the distribution of the species; dc = maximum distance between the centroids; ► indicates the merger of areas.

AoE →	KLR	Area	KIL	Area	KAR	Area	DKM-KWN	Area	MP-LI	Area	DKM-KWN + MP-LI	Area	WCP	Area	AMM	Area	BER	Area	ANR	Area	BER + ANR	Area
dd-dc ↓	syn	km^2^	syn (tot)	km^2^	syn (tot)	km^2^	syn (tot)	km^2^	syn (tot)	km^2^	syn (tot)	km^2^	syn (tot)	km^2^	syn (tot)	km^2^	syn (tot)	km^2^	syn (tot)	km^2^	syn (tot)	km^2^
100–100	<5	/	<5	/	6 (26)	44,295	10 (53)	102,354	11 (55)	108,354	ꟷ	/	21 (57)	146,352	6 (12)	43,465	10 (27)	91,580	10 (34)	85,126	ꟷ	/
300–100	<5	/	<5	/	6 (26)	44,295	13 (53)	114,482	11 (55)	108,354	ꟷ	/	32 (67)	197,269	6 (12)	43,465	►		►		35 (60)	231,762
500–100	<5	/	<5	/	6 (26)	44,295	16 (54)	143,527	11 (55)	108,354	ꟷ	/	35 (67)	197,269	6 (12)	43,465	►		►		40 (61)	242,557
100–150	5 (31)	80,810	8 (35)	127,636	6 (26)	44,295	13 (58)	151,118	14 (59)	148,547	ꟷ	/	25 (67)	190,544	6 (12)	43,465	►		►		25 (60)	224,984
300–150	5 (31)	80,810	11 (36)	144,800	6 (26)	44,295	17 (58)	175,819	14 (59)	148,547	ꟷ	/	32 (67)	197,269	6 (12)	43,465	►		►		35 (60)	231,762
500–150	6 (31)	82,179	17 (39)	221,626	6 (26)	44,295	►		►		38 (94)	379,608	35 (67)	197,269	6 (12)	43,465	►		►		40 (61)	242,557
100–200	5 (31)	80,810	9 (38)	154,709	7 (28)	71,240	►		►		34 (97)	389,194	25 (67)	190,544	6 (12)	43,465	►		►		25 (60)	224,984
300–200	7 (42)	112,281	15 (39)	218,954	7 (28)	71,240	►		►		37 (97)	393,318	32 (67)	197,269	6 (12)	43,465	►		►		35 (60)	231,762
500–200	7 (42)	112,281	17 (39)	221,626	7 (28)	71,240	►		►		41 (97)	419,098	35 (67)	197,269	6 (12)	43,465	►		►		40 (61)	242,557

**Table 2 insects-12-01115-t002:** Maximum number of sinendemisms for AoE by GIE and DBSCAN. Please refer to paragraph 2.2 for the names in full of Areas of Endemism reported in the first row.

	KLR	KIL	KAR	DKM-KWN	MP-LI	DKM-KWN + MP-LI	WCP	AMM	BER + ANR	AMM + BER + ANR
GIE	syn	syn	syn	syn	syn	syn	syn	syn	syn	syn
100 km	/	/	6	/	13	/	20	6	28	/
150 km	5	/	6	17	13	/	31	6	35	/
200 km	5	12	7	19	14	/	32	/	/	51
DBSCAN										
100 km	/	/	6	16	11	/	35	6	40	/
150 km	6	17	6	/	/	38	35	6	40	/
200 km	7	17	7	/	/	41	35	6	40	/

**Table 3 insects-12-01115-t003:** Kendall rank correlation index between the maps obtained by the DBSCAN analysis. dd = maximum diameter of the distribution of the species; dc = distance between the centroids.

	DBSCAN								
dd-dc	100 km/100 km	300 km/100 km	500 km/100 km	100 km/150 km	300 km/150 km	500 km/150 km	100 km/200 km	300 km/200 km	500 km/200 km
100 km/100 km	1	0.522	0.459	1.000	0.522	0.459	1.000	0.522	0.459
300 km/100 km	0.522	1	0.920	0.522	1.000	0.920	0.522	1.000	0.920
500 km/100 km	0.459	0.920	1	0.459	0.920	1.000	0.459	0.920	1.000
100 km/150 km	1.000	0.522	0.459	1	0.522	0.459	1.000	0.522	0.459
300 km/150 km	0.522	1.000	0.920	0.522	1	0.920	0.522	1.000	0.920
500 km/150 km	0.459	0.920	1.000	0.459	0.920	1	0.459	0.920	1.000
100 km/200 km	1.000	0.522	0.459	1.000	0.522	0.459	1	0.522	0.459
300 km/200 km	0.522	1.000	0.920	0.522	1.000	0.920	0.522	1	0.920
500 km/200 km	0.459	0.920	1.000	0.459	0.920	1.000	0.459	0.920	1

## Data Availability

All other data not available in Appendix A can be requested to the authors.

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
