# Peer review of "DBSCAN and GIE, Two Density-Based “Grid-Free” Methods for Finding Areas of Endemism: A Case Study of Flea Beetles (Coleoptera, Chrysomelidae) in the Afrotropical Region"

_insects, 2021, doi:10.3390/insects12121115_

Round 1

Reviewer 1 Report

The paper is well written and organized, the attention of the reader does not wander and the paper is fine except the following minor changes to be done:

  • in GIE what is the Gaussian function for each of occurrence and what is the kernel density function that calculates this occurrence? Give these details for completeness

- give the pseudocode of the algorithm whose steps are given at page 4

- figure 1 is too confuse: maybe a representation as points would be better

- it is unclear what is the technology preferable between DBSCAN and GIE. Reading the conclusion it seems that  DBSCAN is preferable? Or there is some situation where is preferable GIE?

Author Response

The paper is well written and organized, the attention of the reader does not wander and the paper is fine except the following minor changes to be done:

in GIE what is the Gaussian function for each of occurrence and what is the kernel density function that calculates this occurrence? Give these details for completeness

Corrected, we added this information in L 81-85.

- give the pseudocode of the algorithm whose steps are given at page 4

Corrected, we moved the detailed scheme on which the CLUENDA tool is developed from Supplementary Material to the main text (Figure 2).

- figure 1 is too confuse: maybe a representation as points would be better

Corrected.

- it is unclear what is the technology preferable between DBSCAN and GIE. Reading the conclusion it seems that  DBSCAN is preferable? Or there is some situation where is preferable GIE?

Thank you for your comment: we realized that this concept needed more detailed discussion. We added in L 424-433 a paragraph discussing features of both methods. In fact, there is no “final” algorithm, and everything depends on the specific context; we clarified this in the lines reported above.

Reviewer 2 Report

A grid-free algorithm based on the well-known density-based clustering algorithm DBSCAN is applied for  locate areas of endemism for flea beetles in the Afrotropical Region.

This research is interesting, but the authors need to clarify some aspects that are significant in evaluating the performance of the proposed method. Many clustering algorithm-based approaches are used to detect areas on the map where a specific phenomenon is most intense. For example, in the literature there are numerous clustering algorithms proposed to detect such areas on the map (for example, variants of K-means, Fuzzy Cmeans and kernel density clustering algorithms used in hotspot detection). Authors should highlight the performance benefits of using DBSCAN over other clustering algorithms. 

It is necessary to add in section 2 an architectural scheme and a formal schematization, for example in pseudocode, of the proposed method.

Table 1 is hard to read; perhaps it is more useful to split it into two separate tables. 

What are the execution times? It is important, above all, to highlight how execution times vary as the size of the dataset varies and whether the proposed method remains effective even for dealing with massive datasets.

What are the future prospects connected to an improvement in the performances of the proposed method? The text lacks comparative tests with other cluster-based methods proposed in the literature; do the authors intend to make these comparisons in the future? 

Author Response

A grid-free algorithm based on the well-known density-based clustering algorithm DBSCAN is applied for  locate areas of endemism for flea beetles in the Afrotropical Region.

This research is interesting, but the authors need to clarify some aspects that are significant in evaluating the performance of the proposed method. Many clustering algorithm-based approaches are used to detect areas on the map where a specific phenomenon is most intense. For example, in the literature there are numerous clustering algorithms proposed to detect such areas on the map (for example, variants of K-means, Fuzzy Cmeans and kernel density clustering algorithms used in hotspot detection). Authors should highlight the performance benefits of using DBSCAN over other clustering algorithms.

Thank you for your observation, we realized that some of the concepts we reported in the Introduction were not clear enough. We corrected, we added some specific algorithms in Introduction as an example (L 75), and a whole paragraph to better clarify this issue in L 137-141. Specifically, notwithstanding several clustering algorithms are today available, all but the DBSCAN-based ones require an a-priori number of clusters to run the analysis. Of course, considering that our goal is to infer the number of clusters and their location, based on the endemic species occurrences, these cannot be applied. Moreover, other tools measuring spatial aggregation are all based on the Gi* statistics (from Getis and Ord, 1992. The analysis of spatial association by use of distance statistics. – Geogr. Anal. 24: 189–206), which in turn requires a weight assigned to each feature. In the case of endemic species (and to endemicity in general), assigning weights is not possible, as one endemic species is not more important than another.

It is necessary to add in section 2 an architectural scheme and a formal schematization, for example in pseudocode, of the proposed method.

Corrected, we moved the scheme that our proposed tool follows from the Supplementary material to the main text, also changing the numeration of Figures accordingly.

Table 1 is hard to read; perhaps it is more useful to split it into two separate tables.

Corrected, we changed the orientation of Table 1 to give a better readability.

What are the execution times? It is important, above all, to highlight how execution times vary as the size of the dataset varies and whether the proposed method remains effective even for dealing with massive datasets.

Corrected, we performed a supplementary analysis which deals with the execution times, adding the corresponding text in Methods (L 210-216), Results (L 222-225) and Discussion (L 403-404).

What are the future prospects connected to an improvement in the performances of the proposed method? The text lacks comparative tests with other cluster-based methods proposed in the literature; do the authors intend to make these comparisons in the future?

As also clarified in L 137-141 (first response, above), the other methods are not suitable for the purpose of identifying the areas of endemism.

Reviewer 3 Report

The paper proposes a clustering approach for identifying areas of endemism. The approach is compared to the geographical interpolation approach for the case of flea beetles. 

The topic is interesting and worth investigating. The paper is generally well-written and includes a comprehensive enough literature review. The authors have also created a software tool that is included as a supplementary file, facilitating research reproducibility. The approach is described in sufficient details.

My main concern is related to the fact that the paper does not seem to adequately validate that the results of the proposed approach are correct. For example, the paper makes statements such as ”obtained by GIE are in general less clearly defined than the maps obtained by DBSCAN”. The question would be if the obtained maps are (more) correct?

The headers of Table 1 should be explained better. The same issue is also valid for the other tables. For examples, readers lee familiar with the topic might not understand the meaning of ”dd-dc”.

At line 345, the authors are kindly asked to specify why they have chosen to use the Kendall index.

Author Response

The paper proposes a clustering approach for identifying areas of endemism. The approach is compared to the geographical interpolation approach for the case of flea beetles.

The topic is interesting and worth investigating. The paper is generally well-written and includes a comprehensive enough literature review. The authors have also created a software tool that is included as a supplementary file, facilitating research reproducibility. The approach is described in sufficient details.

My main concern is related to the fact that the paper does not seem to adequately validate that the results of the proposed approach are correct. For example, the paper makes statements such as ”obtained by GIE are in general less clearly defined than the maps obtained by DBSCAN”. The question would be if the obtained maps are (more) correct?

Since there are no certain frameworks of reference for these analyses, it is unrealistic to establish the correctness of one method over another in identifying areas of endemism. The only evaluation that can be made as regards the level of significance of the detected areas is based on the lesser or greater support provided by the data used for their identification.

In this perspective, the AoEs identified by DBSCAN are, for the same surface, more significant of those detected by GIE, as generally they are supported by a greater number of sinendemisms.

Indeed, we changed the word ‘defined’ to ‘delimited’, as following your comment we found that ‘delimited’ is more proper to describe the differences in areas.

The headers of Table 1 should be explained better. The same issue is also valid for the other tables. For examples, readers lee familiar with the topic might not understand the meaning of ”dd-dc”.

Corrected, we added the explanation in Table 1 legend.

At line 345, the authors are kindly asked to specify why they have chosen to use the Kendall index.

Corrected, we added this information.

Round 2

Reviewer 2 Report

The authors took into account all my suggestions, improving the quality of their manuscript. I consider this paper publishable in the current form. 

Reviewer 3 Report

I would like to thank the authors for the changes made.